

# Comparative effectiveness and safety of Angio-Seal and StarClose vascular closure devices: a systematic review and meta-analysis

Kun Lai*, Jingquan Chen, Qiang Tan and Lan Luo*

Affiliated Hospital of North Sichuan Medical College, Nanchong, Sichuan Province, China
* These authors contributed equally to this work.

## ABSTRACT

**Objective:** This systematic review and meta-analysis aims to evaluate the effectiveness and safety of Angio-Seal and StarClose vascular closure devices (VCDs) in achieving hemostasis after interventional surgery.

**Methods:** Randomized controlled trials (RCTs) and observational studies comparing Angio-Seal and StarClose were identified through systematic searches. Data on vascular closure success rate, complication rates, hematoma formation, pseudoaneurysm, arterial occlusion, and surgical intervention were extracted and pooled using a random effects model.

**Results:** Nine studies met the inclusion criteria, total 5,466 patients. The meta-analysis revealed a slight, statistically significant difference in the success rate of vascular closure in favor of Angio-Seal (risk ratio (RR) 1.05, 95% confidence interval (CI) [1.00–1.09], $p = 0.03$). No significant difference was found in the overall complication rate, hematoma formation, pseudoaneurysm, arterial occlusion, or surgical intervention.

**Conclusions:** Angio-Seal demonstrated a slightly higher success rate in vessel closure compared to StarClose. Both devices had a comparable safety profile with no significant differences in major complications.

## INTRODUCTION

Achieving hemostasis after intervention is a cornerstone in the management of patients undergoing endovascular procedures (*Kim et al., 2022*). The traditional standard of manual compression, although widely used, has inherent limitations including prolonged patient immobilization, delayed ambulation, and potential discomfort (*Pang et al., 2022*). With the introduction of vascular closure devices (VCDs), there has been a paradigm shift towards improving patient recovery and optimizing clinical workflows (*Koziarz & Kennedy, 2023*). Among the available VCDs, Angio-Seal and StarClose have emerged as prominent contenders in the field, each with its own mechanism of action and purported benefits (*Doshi et al., 2023*).

Corresponding author
Lan Luo, 1827474732@qq.com

Angio-Seal and StarClose differ significantly in their mechanisms of action, each offering distinct advantages and potential limitations. Angio-Seal utilizes a bioabsorbable anchor and collagen plug system that forms a mechanical seal and promotes natural clot formation at the puncture site (*Barton et al., 2023*). This system is advantageous for its rapid hemostasis and bioabsorbability, which minimizes long-term complications (*Choi et al., 2016*). However, Angio-Seal may be less effective in patients with calcified arteries, where anchor deployment can be more challenging.

In contrast, StarClose employs a nitinol clip that grasps the arterial wall to achieve closure (*Bangalore et al., 2011*). The mechanical nature of the clip provides a secure closure, especially in patients with less vascular elasticity or higher levels of arterial calcification (*Lucatelli et al., 2017*). However, the rigid clip structure may cause discomfort or complications in some patients, particularly during repeat procedures or when there is vessel deformation. Understanding these differences allows clinicians to tailor device selection to the specific needs and anatomical considerations of their patients.

This lack of consensus highlights the need for a rigorous synthesis of evidence to assess the comparative effectiveness and safety of Angio-Seal and StarClose (*Postalian, 2022*). Therefore, this systematic review and meta-analysis aims to elucidate the relative performance of these VCDs (*Çakal et al., 2022*). By bringing together data from randomized controlled trials and observational studies, we aim to provide a comprehensive comparison and answer the crucial question of whether one device is superior to the other in clinical practice (*Essibayi et al., 2021*).

## SURVEY METHODOLOGY

### Study protocol and registration

The systematic review and meta-analysis protocol presented in this article was prepared based on the Preferred Reporting Items for Systematic Reviews and Meta-Analyses (PRISMA) guidelines. Further registration takes place with the International Prospective Register of Systematic Reviews (PROSPERO CRD42024529705).

### Literature search strategy

We identified studies by searching the literature in the PubMed, EMBASE, Cochrane Library and Web of Science without any language or publication status restrictions. This search strategy specifically included the search terms for vascular closure devices used, namely: "Angio-Seal" "StarClose" and "hemostasis" combined with Boolean operators to expand the sensitivity of the search. Additional studies were further sought by manually searching the reference lists of identified studies and relevant reviews.

### Eligibility criteria

We will include randomized controlled trials (RCTs) in adult patients comparing Angio-Seal with StarClose for vascular closure after intervention, as well as observational studies (*Lucatelli et al., 2017*; *Deuling et al., 2008*; *Engelbert et al., 2010*; *Elmasri et al., 2017*;

*Veasey et al., 2019*; *Engelbert et al., 2013*; *Rastan et al., 2015*; *Ierardi et al., 2023*; *Gonen, Hakyemez & Erdogan, 2021*). We exclude non-comparative studies and case reports, editorials, or commentaries that do not report interesting results.

## Data extraction

The two reviewers Jingquan Chen and Qiang Tan independently performed data extraction using a standardized form that included the characteristics of the included studies, outcomes, details of the intervention. Any disagreements between reviewers were resolved by consensus or decision by a third reviewer, Lan Luo.

## Statistical analysis

The data was synthesized and analyzed using Review Manager Version 5.4. Risk ratio (RR) and the 95% confidence intervals (CI) were calculated to express the effect size for dichotomous data. To assess heterogeneity among included trials, we used Chi-squared and Higgins $I^2$ tests. If significant heterogeneity was obtained ($p \leq 0.05$ for Chi-squared test results or $I^2 \geq 50\%$), we used a random-effects model; otherwise, a fixed-effects model was used. Statistical significance was indicated by a $p$ value $< 0.05$. In our meta-analysis, we employed both fixed-effects and random-effects models depending on the characteristics of the included studies. The choice between these models was determined not only by the heterogeneity in the results (measured using $I^2$ statistics) but also by potential differences in the study designs. These differences include: Endpoint definitions: Variations in how the studies defined clinical outcomes (*e.g.*, hemostasis success, complication rates).

When significant variability in these factors was present, we utilized a random-effects model, as it allows for more generalized conclusions across studies with diverse clinical settings. Conversely, when the included studies were consistent in their design, a fixed-effects model was used to provide a more precise estimate of the effect size.

## RESULTS

### Study selection

The initial search strategy yielded a total of 163 records, of which 96 remained after duplicate removal. Upon reviewing the titles and abstracts, 73 records were excluded from the research scope due to irrelevant titles or abstracts. The full texts of 21 articles were reviewed, of which nine qualified for the systematic review and meta-analysis. The study selection process is illustrated in the PRISMA flowchart below (Fig. 1).

### Study characteristics

The current meta-analysis includes a number of studies published between 2007 and 2023, which were rigorously selected based on their focus on the efficacy and safety of different vascular closure devices. The collective dataset comes from nine different studies, as summarised in Table 1, covering a range of sample sizes from 166 to 1,672 participants per study, for a total of 5,466 individuals.

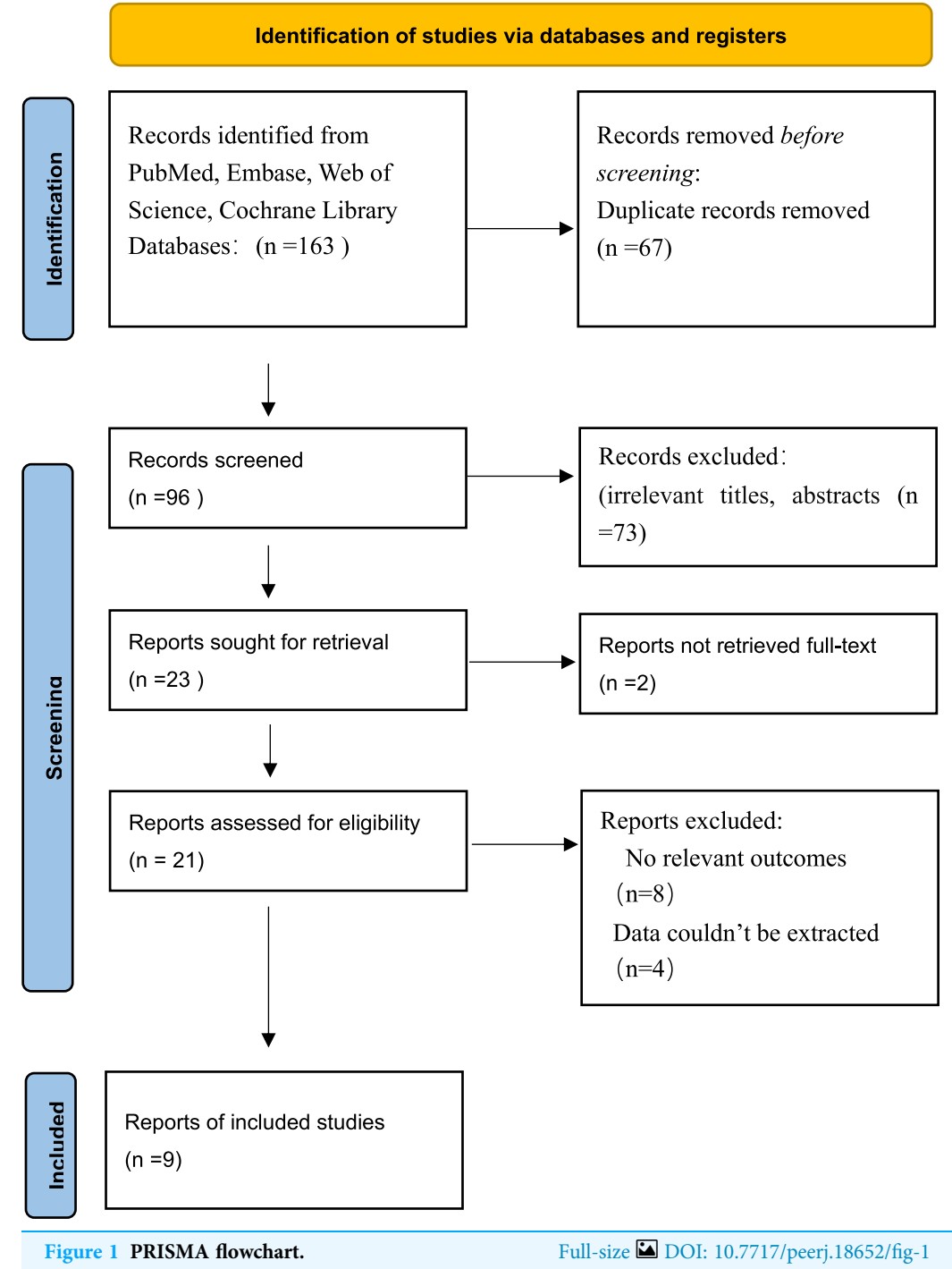

**Figure 1  PRISMA flowchart.**

## Efficacy of vascular closure

The pooled analysis of the success rate of vascular closure indicated a slight advantage for Angio-Seal (RR 1.05, 95% CI [1.00–1.09], $p$ = 0.03). The forest plot for the success rate of vascular closure demonstrates the individual and pooled estimates (Fig. 2).

**Table 1 Study characteristics** (*Elmasri et al., 2017*; *Rastan et al., 2008*; *Ratnam et al., 2007*; *Deuling et al., 2008*; *Veasey et al., 2008*; *Engelbert et al., 2010*; *Lucatelli et al., 2017*; *Gonen, Hakyemez & Erdogan, 2021*; *Ierardi et al., 2023*).

| Study reference | Year | Sample size | Study design | Age (years) | Intervention details |
|---|---|---|---|---|---|
| Matthew A. Elmasri | 2017 | 907 | Retrospective | 59.8 ± 1.09 | AS, FISH, Mynx, perclose, Starclose |
| Aljoscha Rastan | 2008 | 852 | Randomized trial | 67 | Angio-Seal, StarClose, D-Stat Dry |
| Lakshmi A. Ratnam | 2007 | 426 | Prospective nonrandomized | Varying ages | Manual compression, Angio-Seal, StarClose |
| J. H. H. Deuling | 2008 | 450 | Randomized trial | 63.2 ± 11.7 | AS, SC, and manual compression |
| R. A. Veasey | 2008 | 401 | Randomized trial | 65.3 ± 10.6 | AngioSeal, StarClose |
| Travis L. Engelbert | 2010 | 245 | Retrospective | 69.9 ± 9.3 | Angio-Seal, StarClose, D-Stat Dry |
| Pierleone Lucatelli | 2017 | 347 | Prospective | 67 ± 12.3 | Angio-Seal, StarClose |
| Korcan Aysun Gonen | 2021 | 166 | Retrospective | 59.7 ± 12.7 | Angio-Seal, StarClose |
| Anna Maria Ierardi | 2023 | 1,672 | Prospective | 67.5 ± 14.3 | Balloon occluders, non-balloon collagen plug occluders |

**Figure 2 Success rate of vascular closure** (*Elmasri et al., 2017*; *Rastan et al., 2008*; *Ratnam et al., 2007*; *Deuling et al., 2008*; *Veasey et al., 2008*).

## Complication rates

On the other hand, the overall complication rates between these two devices did not reach statistical significance (RR 0.61; 95% CI [0.29–1.29], $p = 0.20$) (Fig. 3).

## Hematomas

The more common complication was hematoma formation, with no statistically significant difference between Angio-Seal and StarClose (RR 0.75, 95% CI [0.46–1.22], $p = 0.24$) (Fig. 4).

## Incidence of pseudoaneurysm

Postprocedure pseudoaneurysm formation is a relatively feared complication. The pooled data for the studies revealed no difference in the incidence of pseudoaneurysms between Angio-Seal and StarClose, with a risk ratio (RR) of 0.52 (95% CI [0.18–1.48], $p = 0.22$) (Fig. 4).

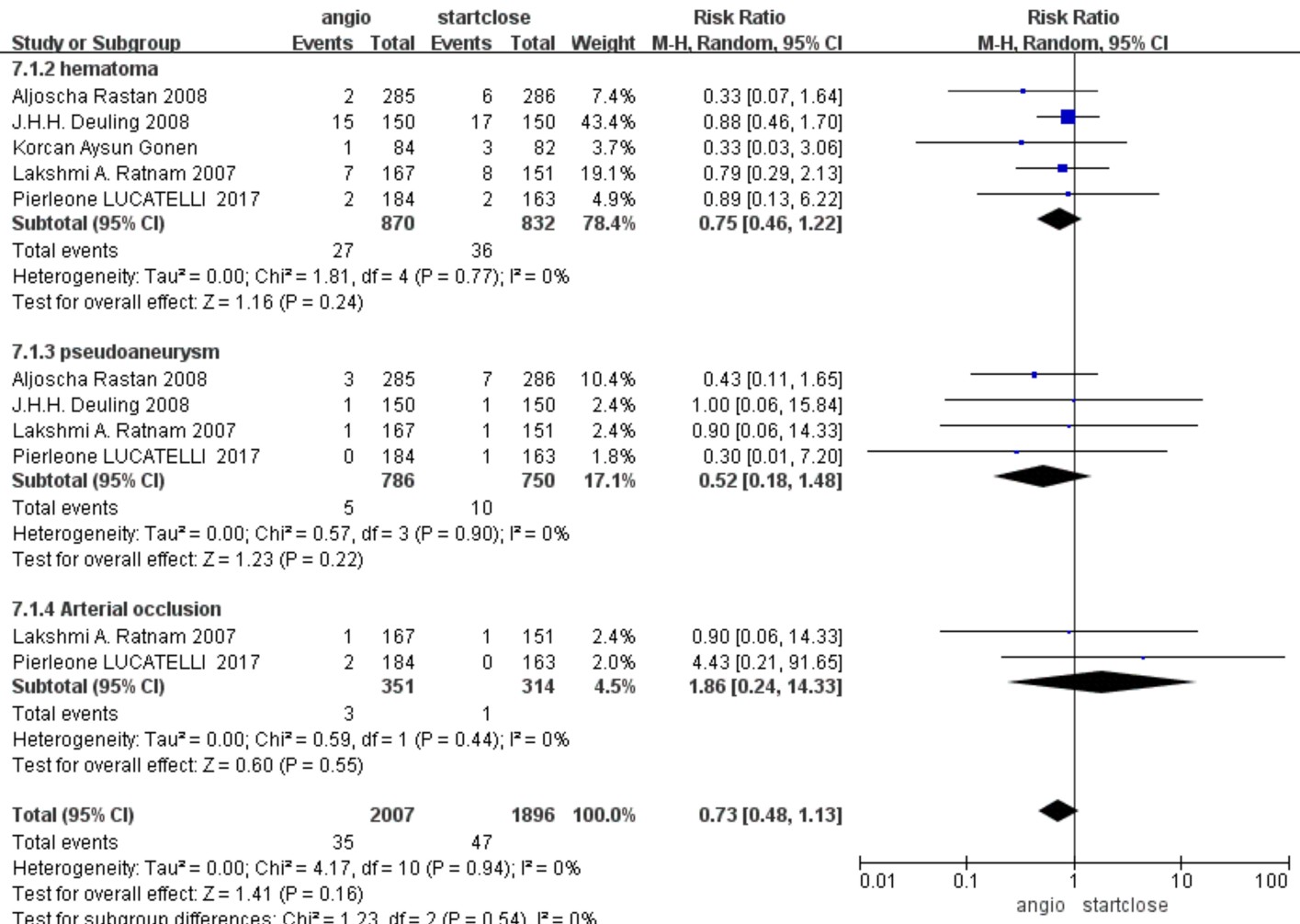

**Figure 3 Complication** (*Elmasri et al., 2017*; *Rastan et al., 2008*; *Ratnam et al., 2007*; *Engelbert et al., 2010*; *Lucatelli et al., 2017*; *Gonen, Hakyemez & Erdogan, 2021*).

**Figure 4 Subgroup complication** (*Rastan et al., 2008*; *Ratnam et al., 2007*; *Deuling et al., 2008*; *Lucatelli et al., 2017*; *Gonen, Hakyemez & Erdogan, 2021*).

**Figure 5 Surgical intervention requirement (*Ratnam et al., 2007*; *Lucatelli et al., 2017*).**

## Arterial occlusion rates

Arterial occlusion is an adverse event, the severity of which can easily lead to harm to the patient. Based on our analysis, it was found that the prevalence of arterial occlusion rates was quite low and the difference between the two groups was not statistically significant, with a relative risk of 1.86 (95% CI [0.24–14.33], $p = 0.55$) (Fig. 4).

## Surgical intervention requirement

It probed into the need for surgical intervention as an adjunct to severe complications related to devices. There was no significant difference in the need for surgical intervention, RR = 1.17 (95% CI [0.13–10.77], $p = 0.89$), in their comparison of the Angio-Seal-treated group and patients treated with StarClose (Fig. 5).

## DISCUSSION

This review aimed at consolidating the current evidence about the efficacy and safety aspects of the two most common vascular closure devices (Angio-Seal and StarClose) by conducting a meta-analysis of aggregate data from a number of comparative studies. This was done to derive pooled effect estimates that would provide health service providers with stronger evidence about the performance of the respective devices in clinical settings.

The analysis demonstrated that Angio-Seal was slightly more effective in achieving hemostasis compared to StarClose, indicated by a modestly higher vascular closure success rate. While this difference is statistically significant, its clinical relevance may vary, requiring attention to individual patient circumstances and procedural details. The slight improvement in the success rate of Angio-Seal suggests its potential advantage in achieving rapid hemostasis, which is particularly important in outpatient procedures. This could lead to earlier ambulation and discharge (*Lucatelli et al., 2017*), ultimately enhancing patient satisfaction and reducing healthcare costs. Clinicians should consider this benefit when selecting a vascular closure device, especially in scenarios requiring swift recovery. The comparable complication rates between Angio-Seal and StarClose highlight a consistent safety profile. This equivalence is essential for clinical decision-making, offering flexibility in choosing a device based on operator experience and patient anatomy without safety limitations.

Timely and effective hemostasis is important for post-procedural care. Angio-Seal's slightly higher success rate may influence clinicians' preference, particularly when immediate hemostasis is vital. The findings of this meta-analysis align with more recent studies, which also suggest that Angio-Seal demonstrates higher technical success and fewer complications compared to StarClose. Recent data reinforces that Angio-Seal remains a reliable choice for achieving immediate hemostasis (*Ierardi et al., 2023*; *Gonen, Hakyemez & Erdogan, 2021*).

## Complications associated with VCDs

There was no significant difference in the incidence of pseudoaneurysm and arterial occlusion between the devices (*Auer et al., 2023*). This observation is consistent with literature indicating that VCDs have improved post-catheterization management but have not eliminated the risk of such complications entirely (*Mohammed & Tamimi, 2017*). These rare complications have significant impacts on patient morbidity, emphasizing the need for vigilant monitoring (*Rastogi et al., 2016*).

Hematoma formation, a frequent complication, occurred at similar rates between the devices, indicating no distinct advantage for either in reducing this risk. This underscores the importance of skilled technique and careful patient selection in the use of VCDs (*Goldsweig & Secemsky, 2020*).

The mechanistic differences between Angio-Seal and StarClose have significant clinical implications. Angio-Seal's bioabsorbable anchor promotes rapid hemostasis by creating a mechanical and biological seal, which may explain its slightly higher success rate in achieving immediate vessel closure. This is particularly advantageous in patients requiring early ambulation or those at higher risk for bleeding. On the other hand, StarClose's nitinol clip provides a mechanical closure that might be more effective in patients with complex vascular anatomies, where a more rigid support structure is needed to ensure closure (*Lucatelli et al., 2017*). Selecting the appropriate device is essential for achieving successful outcomes and minimizing complications. The choice directly influences procedural success and patient recovery.

## Surgical interventions

The rarity of surgical intervention following the use of VCDs and the lack of significant differences between the two devices support the use of VCDs as the primary method of vascular closure following interventional procedures. This is consistent with the primary goal of minimising patient trauma and promoting rapid recovery.

## Limitations

Our study has limitations, including the heterogeneity of the studies, variations in operator expertise, and differences in patient populations, all of which could affect the outcomes.

## Future research

Future research should aim to address several critical gaps in the current literature regarding vascular closure devices. First, large-scale, multicenter randomized controlled trials are needed to evaluate the long-term outcomes of Angio-Seal and StarClose in

diverse patient populations. These studies should focus on specific subgroups, such as those with complex vascular anatomies or high bleeding risks, to determine the most effective device for varying clinical scenarios. Additionally, there is a need for research that investigates the economic implications of using different vascular closure devices. Understanding the cost-effectiveness and impact on healthcare resource utilization will help inform clinical decision-making.

## CONCLUSIONS

In conclusion, this meta-analysis indicates a slight advantage of Angio-Seal in the success rate of vascular closure, maintaining safety comparable to StarClose. However, the selection of a VCD should be customized to the patient's clinical situation, considering the procedural context and operator proficiency.

## ACKNOWLEDGEMENTS

We would like to express our sincere gratitude to all those who contributed to the success of this study. We acknowledge the technical assistance and access to resources from the scientific research development of affiliated hospital of North Sichuan medical college without which this research would not have been possible.

### Funding
The authors received no funding for this work.

### Competing Interests
The authors declare that they have no competing interests.

### Author Contributions
- Kun Lai conceived and designed the experiments, analyzed the data, prepared figures and/or tables, and approved the final draft.
- Jingquan Chen conceived and designed the experiments, performed the experiments, analyzed the data, prepared figures and/or tables, and approved the final draft.
- Qiang Tan performed the experiments, analyzed the data, prepared figures and/or tables, and approved the final draft.
- Lan Luo conceived and designed the experiments, performed the experiments, authored or reviewed drafts of the article, and approved the final draft.

### Data Availability
This is a systematic review/meta-analysis.

### Supplemental Information
Supplemental information for this article can be found online at http://dx.doi.org/10.7717/peerj.18652#supplemental-information.

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
