# Peer review of "Comparative effectiveness and safety of Angio-Seal and StarClose vascular closure devices: a systematic review and meta-analysis"

_PeerJ, doi:10.7717/peerj.18652_

## Round 0.1 · original submission · Major Revisions

As noted by Reviewer 1, it is important to include more recent publications in the analysis.

Reviewer 1 ·

Basic reporting

no comment

Experimental design

The authors reviewed the studies published 2007-2017. These are too old. The authors should review more recent studies.

Validity of the findings

The discussion section is too short. The authors should discuss more. For example,
- What's the difference of mechanism between the 2 devices and how the differences affect outcomes?
- How we can improve the outcomes?
- How we can avoid complications?

Reviewer 2 ·

Basic reporting

The manuscript is written in clear English. The terminology used is appropriate for the target audience. The abstract succinctly summarizes the research, objectives, methods, results, and conclusions.The literature is well-referenced and relevant. Key studies and existing literature on vascular closure devices are cited appropriately.
However, providing more detailed information in the introduction about the differences between the Angio-Seal and StarClose devices, as well as exploring existing literature and potential different applications in the discussion, would enhance the context.

Experimental design

Adherence to PRISMA guidelines and registration in PROSPERO highlight the commitment to transparency and a pre-specified methodological protocol. The inclusion and exclusion criteria for the studies are clearly defined. The search strategy is comprehensive, covering multiple databases without language or publication status restrictions. Methods are described in detail, allowing for replication.
The choice between random-effects and fixed-effects models should be based on potential differences in study designs, such as included patients, treatments, endpoint definitions, or length of follow-up, rather than solely on the heterogeneity of the results. This approach ensures a more accurate and reliable synthesis of the data.Therefore, it would be beneficial for the authors to clarify this aspect in their methodology section.

Validity of the findings

The statistical analysis appears controlled. The results are clearly presented. The slight advantage of Angio-Seal in success rate and the comparable safety profile of both devices are well-supported by the data. However, it's important to note that while this advantage is statistically significant, its practical clinical impact may be minimal. The slight improvement in success rate might not translate into substantial clinical benefits, depending on individual patient circumstances, procedural specifics and operator's expertise.

Additional comments

- Methodologically robust systematic review and meta-analysis
- Clear presentation of the results
- The choice between random-effects and fixed-effects models should take into account potential differences in study designs, such as included patients, treatments, endpoint definitions, or length of follow-up. Please clarify in the methods section
- The introduction could provide more detailed background on the mechanisms of actions, differences, benefits and weakness of the two devices
- The discussion could benefit from more in-depth analysis of potential clinical implications and future research avenues

---

## Round 0.2 · Minor Revisions

Please address the remaining minor revisions requested by Reviewer 2

Reviewer 1 ·

Basic reporting

no comment

Experimental design

no comment

Validity of the findings

no comment

Additional comments

The authors have revised the manuscript following reviewer's comments. I think this manuscript can be published in this version.

Reviewer 2 ·

Basic reporting

no comment

Experimental design

no comment

Validity of the findings

no comment

Additional comments

I commend the authors for their efforts in revising the manuscript.

The revisions have improved the manuscript. The additional details in the introduction, about the differences between Angio-Seal and StarClose, provide helpful context.
However, the discussion, while more detailed, feels repetitive and could be improved:

- Please better elaborate on the clinical relevance of the results (lines 137-138).
- Avoid excessive repetition of words like "crucial" (lines 139, 142, 166) and "critical" (lines 144, 170).
- Include references for the text in lines 160-163
- Consider rewriting the text in lines 164-174 to avoid redundancy.
- Please remove the section from lines 184 to 189, as it is repetitive and redundant with the discussion. The future research section should be focused on how to address the remaining gaps in the literature.

---

## Round 0.3 · accepted · Accept

Dear Dr Lai

Congratulations on the effort of the entire team. The latest revisions were favorable for publication, so your manuscript is accepted.
Sincerely

Reviewer 2 ·

Basic reporting

No comment

Experimental design

No comment

Validity of the findings

No comment

Additional comments

I find the revised version of the manuscript to be cleaner and clearer. I congratulate the authors on their efforts to enhance the manuscript’s quality by implementing the suggested modifications. I have no additional comments.